# Enhanced Luminescent Detection of Circulating Tumor Cells by a 3D Printed Immunomagnetic Concentrator

**DOI:** 10.3390/bios11080278

**Published:** 2021-08-17

**Authors:** Chanyong Park, Abdurhaman Teyib Abafogi, Dinesh Veeran Ponnuvelu, Ilchan Song, Kisung Ko, Sungsu Park

**Affiliations:** 1Department of Medical Device, Korea Institute of Machinery & Materials (KIMM), Daegu 42994, Korea; cksdyd6348@skku.edu; 2School of Mechanical Engineering, Sungkyunkwan University (SKKU), Suwon 16419, Korea; abditeyf@skku.edu (A.T.A.); dinesh@skku.edu (D.V.P.); 3Department of Medicine, College of Medicine, Chung-Ang University, Seoul 06974, Korea; icsong82@cau.ac.kr (I.S.); ksko@cau.ac.kr (K.K.); 4Department of Biomedical Engineering, Sungkyunkwan University (SKKU), Suwon 16419, Korea; 5Institute of Quantum Biophysics (IQB), Sungkyunkwan University (SKKU), Suwon 16419, Korea

**Keywords:** 3D printing, circulating tumor cells, immunomagnetic separation, ATP luminescence assay

## Abstract

Circulating tumor cells (CTCs) are an indicator of metastatic progression and relapse. Since non-CTC cells such as red blood cells outnumber CTCs in the blood, the separation and enrichment of CTCs is key to improving their detection sensitivity. The ATP luminescence assay can measure intracellular ATP to detect cells quickly but has not yet been used for CTC detection in the blood because extracellular ATP in the blood, derived from non-CTCs, interferes with the measurement. Herein, we report on the improvement of the ATP luminescence assay for the detection of CTCs by separating and concentrating CTCs in the blood using a 3D printed immunomagnetic concentrator (3DPIC). Because of its high-aspect-ratio structure and resistance to high flow rates, 3DPIC allows cancer cells in 10 mL to be concentrated 100 times within minutes. This enables the ATP luminescence assay to detect as low as 10 cells in blood, thereby being about 10 times more sensitive than when commercial kits are used for CTC concentration. This is the first time that the ATP luminescence assay was used for the detection of cancer cells in blood. These results demonstrate the feasibility of 3DPIC as a concentrator to improve the detection limit of the ATP luminescence assay for the detection of CTCs.

## 1. Introduction

Circulating tumor cells (CTCs) are cancer cells that are present in the bloodstream after being released from original or metastatic tumors [1]. Since CTCs are found even in the peripheral blood of patients with early-stage cancer [2], they can be used for the early diagnosis of cancer as well as for the prediction of chemotherapy efficacy and cancer relapse [3]. However, these diagnoses based on CTCs are often challenged by the rarity (1–10^2^ CTCs per 1 mL of blood) as well as the heterogeneity of CTCs [4]. To overcome these challenges, various CTC isolation methods have been developed such as density gradient centrifugation [5], microfiltration [6], inertial focusing [7], and immunomagnetic separation (IMS). Among them, IMS, which relies on antibody (Ab)-conjugated magnetic nanoparticles (MNPs), is the most widely used because of its relatively better reliability and reproducibility in isolating CTCs compared to other isolation methods [8]. It can be used to separate CTCs directly from the blood or indirectly by removing non-CTCs, such as blood cells, from the blood [9]. The direct separation strategy has the advantage of allowing the isolation of high-purity CTCs [10,11] but depends highly on the expression of CTC markers, which makes it less favored than other efficient methods [12,13]. The indirect separation strategy has the advantage of allowing the separation of most of the CTCs present in blood, but yields a low level of CTC purity [14].

Much effort has been focused on the development of microfluidic devices (μFDs) for in vitro diagnostics because μFDs are known to be highly useful for the enrichment, isolation, and detection of target cells by allowing continuous fluid manipulation [15,16]. However, conventional μFDs with microchannels (~1 mm) fabricated by photo and/or soft lithography are not suited to the case of diagnostics dealing with large volumes of samples such as 10 mL or more [17]. Three-dimensional (3D) printing is an alternative solution for fabricating microchannels larger than 1 mm and, more recently, 3D printed μFDs have been used for the isolation of bacteria and animal cells [18,19]. However, no reports have yet been published that utilize 3D printed μFDs as an effective device for the isolation of CTCs.

The adenosine triphosphate (ATP) luminescence assay is the conventionally employed method to estimate intracellular ATP, which is the primary energy unit of living cells [20,21]. ATP oxidizes luciferin, and the oxidized luciferin emits light at 580 nm. The light is easily detected using a luminometer. The sensitivity of a luminometer is high enough (~10–11 M) to monitor ATP release in cell suspensions and tissue preparations [22]. Because of its high sensitivity and simplicity, it has been widely used for the detection of various types of cells [19,23], the measurement of the chemosensitivity of cancer cells, and the estimation of cell activity [24]. However, it has not been used for the detection of CTCs in blood, because ATP in blood interferes with CTC detection.

In this study, we report on a sensitive and rapid method to isolate and simultaneously detect CTCs in blood using a 3D printed immunomagnetic concentrator (3DPIC) and the ATP luminescence assay, a procedure requiring a total of 30 min, including 20 min of incubation time (Figure 1). MNPs (average diameter, 50 nm) were conjugated with a monoclonal antibody (mAb) targeted to epithelial cell adhesion molecule (EpCAM), one of the CTC biomarkers [25]. Colon and breast cancer cells were first enriched in 3DPIC through IMS and later enumerated using the ATP luminescence assay. The specificity of the enriched cells was verified using immunostaining. To demonstrate its feasibility for isolation and detection of CTCs in blood, either colon or breast cancer cells in 1 mL of blood were enriched using 3DPIC at 5 mL/min. This allowed the ATP luminescence assay to detect as little as 10 cells in the blood within several minutes. This is the first time that the ATP luminescence assay is used for the detection of cancer cells in blood.

## 2. Materials and Methods

### 2.1. Cell Culture

The colon cancer cell Caco-2 and breast cancer cell MCF-7 lines were purchased from the American Type Culture Collection (ATCC) (Manassas, VA, USA). Caco-2 and MCF-7 cells were cultured in Dulbecco’s Modified Eagle Media (DMEM) (Global life science solutions, Marlborough, MA, USA) supplemented with 10% (*v*/*v*) fetal bovine serum (FBS) (HyClone Laboratories, Inc., Logan, UT, USA), 100 units/mL of penicillin (Life Technologies, Carlsbad, CA, USA), and 100 μg/mL of streptomycin (Life Technologies, Rockville, MD, USA). The cells were maintained at 37 °C with 5% CO_2_ and 95% relative humidity.

### 2.2. Conjugation of the Antibody to MNPs

Amine-modified MNPs with an iron oxide (Fe_3_O_4_) core (fluidMAG-Amine, 50 nm in diameter) were purchased from Chemicell Co. (Berlin, Germany). mAb CO17-1A from transgenic tobacco plants, which targets EpCAM, was provided by the lab of K. Ko at Chung-Ang University, Korea [26]. Its binding affinity to EpCAM was proved to be superior to that of the commercial EpCAM antibody (R&D Systems, Minneapolis, MN, USA) by surface plasmon resonance (SPR) (ProteOn XPR36; Bio-Rad, Hercules, CA, USA) (Appendix A).

MNPs were sonicated for about 40 s to reduce aggregation. A solution of MNPs (1 mg/mL, 10^10^ particles/mL, final conc.) in phosphate-buffered saline (PBS, pH 7.4) was mixed with glutaraldehyde (2.5%, *v*/*v*) in PBS at room temperature (RT) for 1 h using a rotary incubator as previously reported [27].

### 2.3. Fabrication of 3DPIC

The 3DPIC (Figure 2a) was designed to have a channel (3.2 mm in diameter) and a cylindrical hole (20 mm in diameter) using the Student version of Inventor^®^ Professional (Autodesk Inc., Seoul, Korea) [18]. Its 3D model was sectioned along the z-axis and converted to a compatible image file for a digital light processing (DLP) 3D printer (IM-96, Carima Co., Seoul, Korea). Photocurable polymer urethane (CUB035C; Carima Co.) was exposed to UV light at 405 nm following the pattern of each image file, thus fabricating 3DPIC layer by layer. The printout (Figure 2b) was washed with 70% ethanol to remove the residual polymer in the channels and then dried at RT for 2 min. It was further exposed to UV light at 405 nm to further increase its strength. The 3DPIC consists of a channel with a diameter of 3.2 mm to hold the permanent magnet (diameter, 20 mm), as shown in Figure 2a,b.

### 2.4. Enrichment of CTCs in Buffer and Blood Using 3DPIC

Two hundred microliters of mAb-MNPs (10^10^ particles/mL, final conc.) was mixed with 10 mL of PBS containing freshly cultured cancer cells (1–10^4^ cells/mL, final conc.) and incubated at 37 °C on an orbital shaker at 200 rpm for 20 min. The mixture was then transferred to a syringe, and a syringe pump (Harvard Apparatus, Boston, MA, USA) was then used to circulate the mixture at different flow rates (1–100 mL/min) into 3DPIC while a permanent magnet (Ø 20 mm × 20 mm, 527 mT) was plugged on 3DPIC (Figure 2b). Once CTCs–mAb–MNPs complexes were enriched in 3DPIC, the enriched complexes in the channel were washed two times with about 400 µL of PBS. Finally, the permanent magnet was removed, and CTCs–mAb–MNPs complexes were collected into tubes through the outlet port by flowing 100 µL of PBS into 3DPIC (Figure 2a).

Whole blood treated with 0.1% K2 ethylenediaminetetraacetic acid (EDTA) was purchased from Innovative Research, Inc. (Novi, MI, USA). The hematocrit (hct) value was about 39%. Blood was spiked with cancer cells at different concentrations (1–10^4^ cells/mL, final conc.). Then, 1 mL of the spiked blood was mixed with 9 mL of PBS containing Ab–MNPs (10^10^ particles/mL, final conc.). All the following steps for the enrichment were similar to those mentioned above.

### 2.5. Detection of CTCs Using te ATP Luminescence Assay

After enrichment in 3DPIC, the enriched CTCs were collected and incubated at 95 ℃ for 10 min to release ATP. The samples were then transferred into a LuciPac Pen (Kikkoman Biochemifa Co., Tokyo, Japan) containing luciferase. ATP luminescence intensity was measured according to the protocol provided by the manufacturer. In detail, the LuciPac Pen was inserted into the Lumitester PD-30 (Kikkoman Biochemifa Co.), and ATP luminescence intensity was measured within 30 s. The amount of ATP present in the sample was quantified by the amount of light emitted during the measurement and is described in relative light units (RLU).

### 2.6. Calculation of the Capturing Efficiency

The capturing efficiency was evaluated by comparing the luminescence intensity of the captured cells with the intensity of the original cells. To calculate the capturing efficiency, we evaluated the luminescence intensity at various cell concentrations (1–10^4^ cell/mL) and created a calibration curve. Finally, the cell number of the enriched sample was determined by interpolating the luminescence intensity obtained by the ATP luminescence assay with the data on the calibration curve. The capturing efficiency was calculated using the following formula [19].
Capturing efficiency (%) = Ne/Nt × 100 %
where Nt and Ne are the number of cells in a sample and the number of captured cells in the sample, respectively.

### 2.7. Immunostaining of CTCs and Blood Cells

The enriched CTCs in an isolated sample and blood cells in the eluent were carefully washed with PBS, fixed in 4% paraformaldehyde for 15 min at RT, and permeabilized with PBS containing 0.15% (*v*/*v*) Triton X-100 (Sigma–Aldrich, St. Louis, MO, USA) at RT for 20 min. They were then blocked with 3% BSA at room temperature (RT) for 1 h. The samples were incubated overnight with Alexa Fluor 594-conjugated human CD45 antibody (R&D Systems, Inc., Minneapolis, MN, USA) and Alexa Fluor 488-conjugated mouse EpCAM (Thermo Fisher Scientific) at 4 °C. Then, the nuclei of the cells were stained with 4′,6-diamidino-2-phenylindole (DAPI) (Sigma–Aldrich). Images were taken under an epi-fluorescence microscope (DeltaVision^®^ Elite, GE Healthcare, Chicago, IL, USA).

### 2.8. Statistical Data Analysis

Data representation was based on the mean ± standard deviation of three or more independent experiments. We used Student’s *t*-test to compare data under various conditions.

## 3. Results and Discussion

### 3.1. Effect of mAb Concentration and Flow Rate on the Capturing Efficiency of CTCs by 3DPIC

To find the optimal concentration of mAb for its conjugation with MNPs, the capturing efficiency of Caco-2 cells by MNPs conjugated with the mAb (CO17-1A) at varying concentrations (0.1–20 µg/mL) was measured. At 5 µg/mL of mAb, the capturing efficiency was about 90%, and no significant increment in capturing efficiency was observed when the concentration was increased (Figure 2c), showing that 5 µg/mL of mAb was sufficient to react with 10^10^ particles/mL of MNPs for the isolation of CTCs.

To find the optimal flow rate for the isolation of CTCs, 10 mL of PBS containing Caco-2 cells (10^4^ cells/mL, final conc.) and mAb–MNPs (10^10^ particles/mL, final conc.) were injected into 3DPIC at various flow rates (1–100 mL/min). The capturing efficiencies at 1 and 5 mL/min were found to be about 90% (Figure 2d). At flow rates higher than 5 mL/min, the capturing efficiency significantly decreased. Therefore, 5 mL/min of flow rate was chosen for the isolation of CTCs in 3DPIC. The 3DPIC was comprised of a W-shaped microchannel (Figure 2a,b), which previously showed higher capturing efficiency for bacterial pathogens than a spiral microchannel [18]. Owing to the geometrical characteristic of the W-shaped microchannel, CTCs–mAb–MNPs complexes were likely to slow down before the wide curve in the lateral region where the permanent magnet was located (Figure 1), similar to bacteria–Ab–MNPs complexes [18], thus allowing an easy capturing into the microchannel even at the high flow rates. In addition, 3DPIC was made of plastic and able to withstand the pressure generated at high flow rates, such as 5 mL/min [28]. This explains how CTCs could be efficiently captured in 3DPIC even at the high flow rates (1–5 mL/min).

### 3.2. Improvement in the ATP Luminescence Assay for the Detection of Cancer Cells by 3DPIC

Once CTCs are enriched by 3DPIC, the LODs of the ATP luminescence assay can be lowered. Without the use of 3DPIC, the LOD of the ATP luminescence assay for Caco-2 and MCF-7 in PBS was 10 cells per mL. However, the LOD was lowered to 1 cell per mL when 3DPIC was used to enrich cells from 10 mL of PBS. More remarkably, the bioluminescence signal at all tested concentrations of cells increased about 10 times when 3DPIC was used, which is equivalent to a 10-fold enrichment according to the graphs (Figure 3a,c). This improvement in detection sensitivity by 3DPIC can be explained by high capturing efficiencies (about 80 to 90%) for all the tested cancer cells in 1 to 10^4^ cells/mL, as shown in Figure 3b,d. Intracellular ATP can vary depending on the type of cancer cell, and the application of ATP luminescence assays for the detection of CTC in clinical samples requires more extensive testing with different types of cancer cells.

### 3.3. Comparison between 3DPIC and a Commercial Cell Separation Kit for Cancer Cell Isolation

The improved sensitivity of the ATP luminescence assay by 3DPIC was further evaluated through comparison with a CTC separation kit (EasySEP Human EpCAM-positive selection cocktail). The LOD of 3DPIC for Caco-2 and MCF-7 cells was 1 cell/mL, which is one-order of magnitude lower than that of the kit (10 cells/mL), and the R^2^ of 3DPIC (1.00) was higher than that of the kit (0.96) (Figure 4a,c). This clearly proves the better performance of 3DPIC compared to the kit (Figure 4b,d). The kit was able to process up to 2 mL of sample, while 3DPIC was able to process up to 100 mL. The larger the sample volume, the greater the enrichment reached [16,19], which in turn results in a better LOD of the ATP luminescence analysis. It is known that the average blood volume collected for any liquid biopsy is 7.5 mL [29,30,31]. Therefore, 3DPIC could be advantageous for CTC detection with respect to commercial kits.

### 3.4. Spike Test in Blood

Blood contains ATP molecules that increase the background luminescence intensity during the ATP luminescence assay, even in the absence of CTCs. Because of this, the detection of CTCs at low concentrations is nearly impossible using the ATP luminescence assay. Therefore, the capturing of CTCs using mAb–MNPs in blood and their isolation from other cells should be performed simultaneously. While performing the assay without washing, the color of the eluent was the same as that of the blood, with an optical intensity value of 1000 RLU, indicating the presence of blood cells (Appendix A). After the first washing step, the color of the eluent faded, whereas the optical intensity remained the same, proving the presence of blood cells in the sample volume. This made us perform a second washing, during which the eluent became transparent, with a reduction in optical intensity by one order. After more washings, the eluent became brighter owing to the non-specific binding of blood cells and its removal from 3DPIC. In addition, the blood cells containing ATP were removed, thereby reducing the intensity to a non-detectable limit. Hence, to detect CTCs in blood with the current methodology, it is necessary to wash the blood twice to remove the residual blood cells.

The intensity increased in proportion to the increase in the concentration of CTCs in the blood with high capturing efficiency (Figure 5a,b). The LOD of 3DPIC for CTCs in the blood was 10 cells/mL. The results for the blood samples were not as good as those for the PBS samples. This difference might be due to the non-specific binding of blood cells inside 3DPIC, which increased the intensity in the samples. A similar observation has been reported. Ten CTCs were detected by enriching 1 mL of blood; therefore, the system can be used to detect CTCs in liquid biopsies.

Enriched cells were immunostained to confirm whether CTCs were present in the isolated sample. DAPI, which stains the nuclei of cells, EpCAM, a marker of CTC, and CD45, a marker of blood cells, were used together for immunostaining. In the case of isolated samples, DAPI and EpCAM staining was positive, whereas CD45 staining was negative (Figure 5c). In the case of the eluent, DAPI and CD45 staining was positive, whereas EpCAM staining was negative. This staining clearly confirmed that CTCs with EpCAM could be selectively captured in the blood flow analysis, which paves way for selective CTC detection in blood.

## 4. Conclusions

In this study, we developed the novel 3DPIC with the ATP luminescence assay for the enrichment and rapid detection of CTCs in blood. Taking advantage of the design flexibility afforded by 3D printing, a curve channel inside 3DPIC was fabricated, which is difficult to fabricate using conventional fabrication methods for μFDs, such as soft lithography. The curved channel reduced the velocity of the sample, increasing the capturing efficiency. The 3DPIC based on continuous flow can handle a large volume of sample compared to a commercial kit based on conventional batch processes, which improves the LOD of 3DPIC by more than one order compared to the kit. Various types of CTCs can be enriched and detected with 3DPIC within 30 min, up to 10 cell/mL of CTCs in blood. The ATP luminescence assay, which cannot be applied to blood samples due to the presence of extra ATP in blood, was combined with IMS to selectively capture only CTCs in blood to measure ATPs only from CTCs. Our results demonstrate that 3DPIC with the ATP luminescence assay could be applied in diagnostic fields such as liquid biopsy analysis requiring target isolation and rapid detection.

## Figures and Tables

**Figure 1 biosensors-11-00278-f001:**
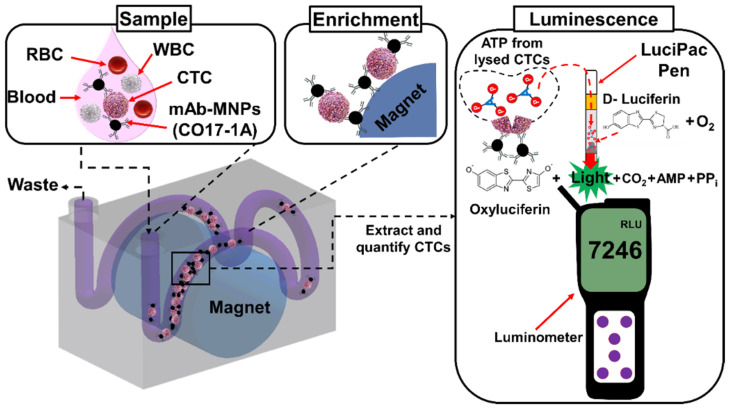
Schematics describing the selective enrichment of CTCs in blood using 3DPIC, followed by their detection using the ATP luminescence assay. CTCs in blood are enriched with MNPs conjugated with a mAb (CO17-1A) specific for EpCAM, using a permanent magnet in 3DPIC. Then, the enriched CTCs are transferred into a LuciPac pen. In the pen, CTCs are lysed, releasing ATP. ATP then oxidizes luciferin in the presence of oxygen (O_2_), producing oxidized luciferin (oxyluciferin) with carbon dioxide (CO_2_), pyrophosphate (PP_i_), and adenosine monophosphate (AMP). Finally, oxyluciferin emits light, and the emitted light is measured by an ATP luminometer. RLU: relative light unit; WBC: white blood cells; RBC: red blood cells.

**Figure 2 biosensors-11-00278-f002:**
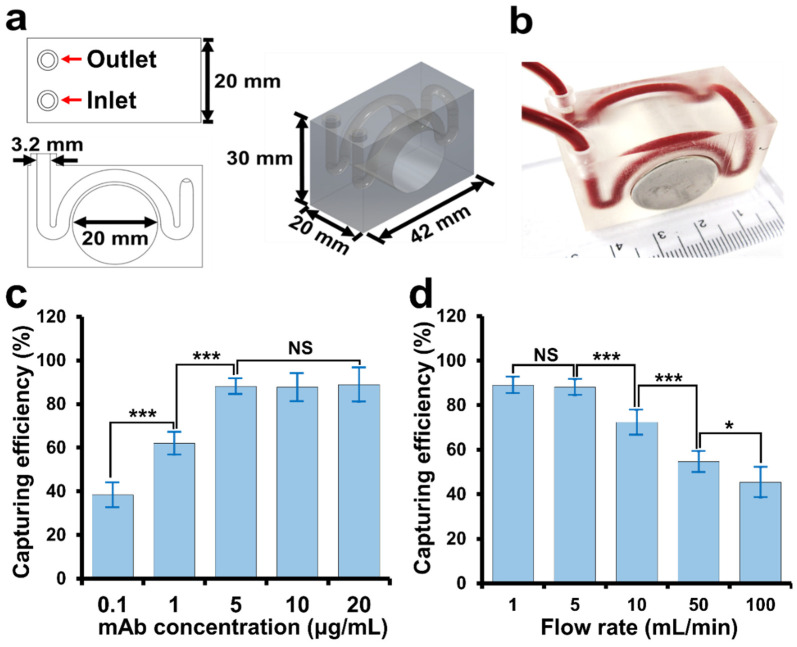
Fabrication and optimization of 3DPIC. (**a**) Design and (**b**) printed image of 3DPIC. The 3DPIC consists of a round channel with a diameter of 3.2 mm and a cylindrical hole holding a permanent magnet (diameter, 20 mm). (**c**) Effect of the concentration of mAb on the capturing efficiency of Caco-2 cells by 3DPIC. Two hundred microliters of MNPs (10^10^ particles/mL, final conc.) that were previously conjugated with mAb at different concentrations (0.1–20 µg/mL) in PBS were mixed with Caco-2 cells (10^4^ cells/mL, final conc.) in 10 mL of PBS and incubated at 37 ℃ on an orbital shaker at 200 rpm for 20 min. Then, the mixture was injected into 3DPIC at 5 mL/min. The captured cells were counted using the ATP luminescence assay. (**d**) Capturing efficiency of CTCs (10^4^ cell/mL) in 10 mL of PBS at different flow rates (1–100 mL/min) with MNPs conjugated with the mAb at 5 µg/mL. Student’s *t*-test, NS: non-significance, *: *p* < 0.05, and ***: *p* < 0.001. Sample number (*n*) = 3.

**Figure 3 biosensors-11-00278-f003:**
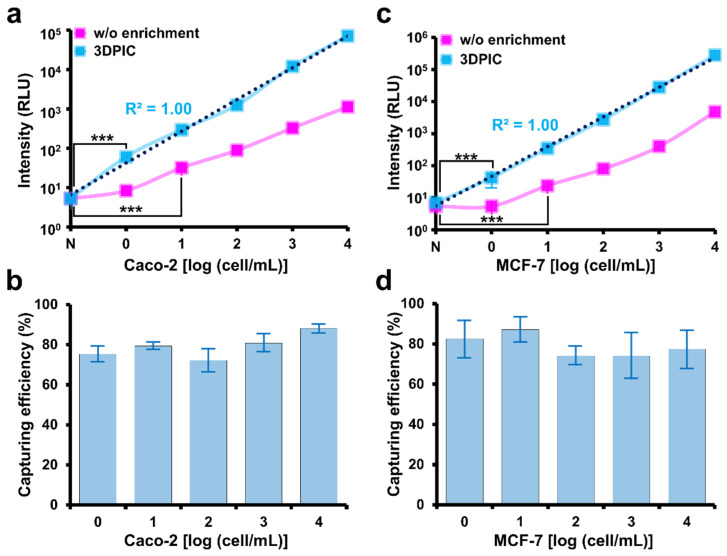
Isolation of cancer cells from PBS by 3DPIC and determination of the number of isolated cancer cells by the ATP luminescence assay. Number of Caco-2 (**a**) and MCF-7 (**c**) cells at different concentrations (1–10^4^ cells/mL, final conc.) in 10 mL of PBS determined by the ATP luminescence assay with and without 3DPIC. Capturing efficiency of Caco-2 (**b**) and MCF-7 (**d**) cells at different concentrations (1–10^4^ cells/mL, final conc.) in 10 mL of PBS by 3DPIC. Student’s *t*-test, *** *p* < 0.001, *n* = 3.

**Figure 4 biosensors-11-00278-f004:**
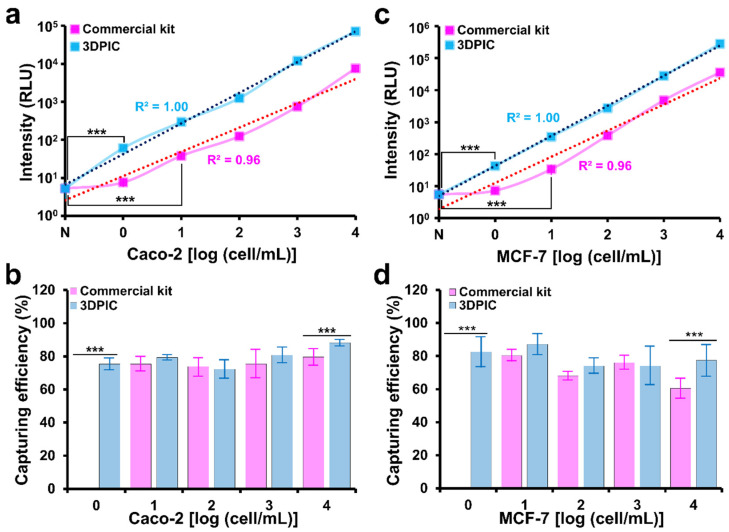
Comparison between 3DPIC and a commercial cell separation kit for cancer cell isolation. Number (**a**,**c**) and capturing efficiency (**b**,**d**) of Caco-2 (**a**,**b**) and MCF-7 (**c**,**d**) cells at different concentrations (1–10^4^ cells/mL, final conc.) isolated by the commercial kit and 3DPIC using the ATP luminescence assay. Student’s *t*-test, *** *p* < 0.001, *n* = 3.

**Figure 5 biosensors-11-00278-f005:**
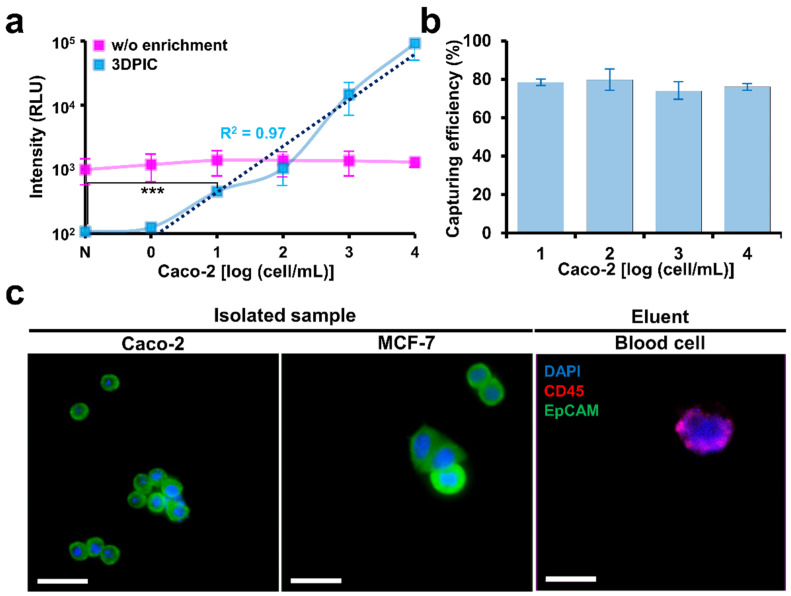
Spike test in blood. (**a**) Luminescence change for CTCs at different concentrations (1–10^4^ cells/mL) in blood in 3DPIC. (**b**) Capturing efficiency at different concentrations of CTCs (1–10^4^ cells/mL) in 1 mL of blood in 3DPIC. (**c**) Immunostaining of CTCs (Caco-2 and MCF-7 cells) and blood cells after capturing and isolation. CTCs were in the isolated sample, and blood cells were in the eluent. The cells were stained using CD45 (red), EpCAM (green), and DAPI (blue). The scale bar is 50 μm. Student’s *t*-test, *** *p* < 0.001, *n* = 3.

## Data Availability

Not applicable.

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
