# Peer review of "Enhanced Luminescent Detection of Circulating Tumor Cells by a 3D Printed Immunomagnetic Concentrator"

_biosensors, 2021, doi:10.3390/bios11080278_

Round 1

Reviewer 1 Report

The authors have presented a novel concentrator. The authors have designed their experiments properly and presented them in an easy to understand language

While I do not have major comments with regard to the scope of the manuscript, The flow geometry they have used is novel and it would be interesting to look at the flow kinetics and its contribution to the capture efficiency.

I recommend the manuscript for publication

Author Response

Thank you!

Reviewer 2 Report

The authors reported a 3D-printed fluidic device for CTC capturing relying on antibody-linked magnetic nanoparticles. They stated that the curved design of the device would increase the capture efficiency. The isolated CTCs were analyzed by an ATP immuno-assay using commercialized kits.

Overall, I think the manuscript is well prepared. However, several issues need to be addressed before considering its publication:

1) The 3D printed curved device is considered one of the major novelties. However, the authors did not show how does the curved channel help in increasing the CTC enrichment performance as compared to conventional designs, for example, a planar channel. The authors should demonstrate it.

2) The size of the device is in centimeters and of the channel is in millimeters. Under such dimensions, I think there are other simple routes to build such a fluidic structure. For example, just roll the soft plastic tubing around a cylindrical magnet. The author should highlight the advantage of using 3D printing for the curved fluidic device fabrication.

3) Some of the statements might be inappropriate or lacking explanation: for instance, line 187 & 282, “CTCs-mAb-MNPs complexes were likely to slow down before the 187 wide curve at the lateral region where the permanent magnet located”, “The curved channel reduced the velocity of the sample”, why?

4) The authors used “for the first time”, “first” for couple of times in the manuscript. I suggest that avoid using this statements.

Reviewer 3 Report

The authors report on a 3D printed immunomagnetic device for isolation and enrichment of circulating tumour cells (CTCs) from blood. They use antibody coated magnetic nanoparticles to sort CTCs from different samples and then detect the concentration of CTCs with an ATP luminescence assay. They compare the limit of detection of their device with the commercially available kit and report an improvement in limit of detection of CTCs which is an order of magnitude higher than the limit of detection in commercially available kits. In my opinion, the paper will be of interest to the readers of biosensors. However, some minor editions could enhance the quality of the manuscript. Please see my comments below:

  1. The authors should comment on how increasing/decreasing channel diameter could lead to a change in device efficiency? Does increasing channel diameter decrease capture efficiently?
  2. what determines the 100 mL upper limit of the device? Why is the device not capable of handling more than 100 mL?
  3. The enrichment and efficiency of detection is improved with this device. Therefore, one would expect that a smaller volume of blood sample compared with the amount required in conventional methods (7.5 mL) would suffice for the test in this device. The authors should comment on this fact.

Round 2

Reviewer 2 Report

no further comments.